# Mirror-induced reflection in the frequency domain

Yaowen Hu [1,2,6] ✉, Mengjie Yu[1,3,6], Neil Sinclair[1,4], Di Zhu [1], Rebecca Cheng[1], Cheng Wang [5] & Marko Lončar [1] ✉

Mirrors are ubiquitous in optics and are used to control the propagation of optical signals in space. Here we propose and demonstrate frequency domain mirrors that provide reflections of the optical energy in a frequency synthetic dimension, using electro-optic modulation. First, we theoretically explore the concept of frequency mirrors with the investigation of propagation loss, and reflectivity in the frequency domain. Next, we explore the mirror formed through polarization mode-splitting in a thin-film lithium niobate micro-resonator. By exciting the Bloch waves of the synthetic frequency crystal with different wave vectors, we show various states formed by the interference between forward propagating and reflected waves. Finally, we expand on this idea, and generate tunable frequency mirrors as well as demonstrate trapped states formed by these mirrors using coupled lithium niobate micro-resonators. The ability to control the flow of light in the frequency domain could enable a wide range of applications, including the study of random walks, boson sampling, frequency comb sources, optical computation, and topological photonics. Furthermore, demonstration of optical elements such as cavities, lasers, and photonic crystals in the frequency domain, may be possible.

Synthetic dimensions, typically formed by a set of atomic[1,2] or optical modes[3–9], allow simulations of complex structures that are hard to do in real space, as well as high-dimensional systems beyond three-dimensional Euclidian space. Therefore, synthetic dimensions provide opportunities to investigate and predict, in a controlled manner, a wide range of physical phenomena occurring in e.g., ultracold atoms, solid state physics, chemistry, biology, and optics[3,10–12]. Exploration of synthetic dimensions using optics has been of particular interest in recent decades, leveraging several of degrees of freedom of light, including space[4–6,13], frequency[14–25], time[9,26], and orbital angular momentum[22,27].

Integrated optics is an ideal platform for creating synthetic dimensions in the frequency domain, due to the high frequency and

bandwidth of light, availability of strong nonlinear interactions, good stability and coherence of the modes, scalability, and excellent reconfigurability[11]. Furthermore, the ability to tailor the gain and loss within an optical system naturally allows the investigation of non-Hermitian physics which are typically hard to explore in other physical systems. Frequency synthetic dimensions in photonics has recently been experimentally investigated, including the measurement of band structure[14] and density of states (DOS) of frequency crystals up to four dimensions[16], realization of two synthetic dimensions in one cavity[7], dynamical band structure measurement[15], topological windings[8] and braiding[28] in non-Hermitian bands, spectral long-range coupling[17], high-dimensional frequency conversion[19], frequency diffraction[24], and Bloch oscillations[25,29,30]. With a few exceptions[16,31], investigations of

[1]John A. Paulson School of Engineering and Applied Sciences, Harvard University, Cambridge, MA 02138, USA. [2]Department of Physics, Harvard University, Cambridge, MA 02138, USA. [3]Ming Hsieh Department of Electrical and Computer Engineering, University of Southern California, Los Angeles, CA 90089, USA. [4]Division of Physics, Mathematics and Astronomy, and Alliance for Quantum Technologies (AQT), California Institute of Technology, Pasadena, CA 91125, USA. [5]Department of Electrical Engineering and State Key Laboratory of Terahertz and Millimeter Waves, City University of Hong Kong, Kowloon, Hong Kong, China. [6]These authors contributed equally: Yaowen Hu, Mengjie Yu. ✉e-mail: yaowenhu@fas.harvard.edu; loncar@seas.harvard.edu

synthetic frequency dimensions on photonic chips have not been extensively studied. In particular, one of the most fundamental phenomena--the reflection of light by synthetic mirrors--has not been investigated yet in frequency synthetic dimensions.

Here we study, both theoretically and experimentally, reflection and interference of optical energy propagating in a discretized frequency space, i.e., a one-dimensional frequency crystal, caused by frequency-domain mirrors introduced in such a frequency crystal. The lattice points of the frequency crystal are formed by a set of frequency modes inside a thin-film lithium niobate (TFLN) microresonator, and the lattice constant is determined by the free spectral range (FSR) of the resonator (for a single spatial mode)[16]. Applying a continuous-wave (CW) electro-optic phase modulation to the optical resonator (Fig. 1a), at a frequency equal to the FSR (microwave-frequency range), results in coupling between adjacent frequency modes. Photons injected into such crystals can hop from one lattice site to another, leading to a tight-binding crystal[11,16]. The coupling strength between nearest neighbor lattice points, $\Omega$, (Fig. 1b) is proportional to the voltage of the microwave driving signal and is related to the conventional modulation index $\beta$ of a phase modulator as $\Omega = \frac{\beta}{2\pi}$FSR (in a conventional modulator, the relationship between $\beta$ and the driving voltage $V$ is $\beta = \pi \frac{V}{V_\pi}$, in which $V_\pi$ is the voltage required to achieve a $\pi$ phase shift)[32]. As a result, when injecting a CW optical signal into one of the crystal lattices sites (cavity resonances), optical energy spreads along the frequency synthetic dimension. A defect introduced in the frequency crystal can break the discrete translational symmetry of the lattice, resulting in reflection of light in the frequency domain (Fig. 1b). The defect can serve as a mirror in the frequency crystal, which is the frequency analog of a mirror in real space (Fig. 1c). The frequency mirror can be introduced by a mode splitting that is induced by coupling specific lattice points to additional frequency modes (Fig. 1d). These additional modes can be different spatial or polarization modes, clockwise and counterclockwise propagating modes of a cavity, or modes provided by

additional cavities. In this work, we first use coupling between the traverse-magnetic (TM) and traverse-electrical (TE) modes to realize mirrors for the latter. Then, we show that the frequency mirrors can also be realized using coupled resonators, an approach that allows better control, reconfigurability, and is more tolerant to fabrication imperfections.

## Results

### Theory

We first theoretically investigate the frequency crystal dynamics for a reflection. For a conventional electro-optic frequency crystal without mirrors[16], the Hamiltonian is described by $H = \sum_{j=-N}^{N} \left( \omega_j a_j^\dagger a_j + \Omega \cos \omega_m t \left( a_j^\dagger a_{j+1} + h.c. \right) \right)$ where $a_j$ represents each frequency mode and $\omega_m$ is the modulation frequency that equal to the FSR. In the rotating frame of each mode $a_j \rightarrow a_j e^{-i(\omega_L + j\omega_m)t}$, they are all frequency-degenerate with a tight-binding coupling, i.e., $H = \sum_{j=-N}^{N} \left( \frac{\Omega}{2} \left( a_j^\dagger a_{j+1} + h.c. \right) \right)$. As a result, Lorentzian resonances of the resonator are broadened and have a profile corresponding to the DOS of the crystal (Fig. 2a)[16]. Therefore, varying the laser detuning $\Delta = \omega_L - \omega_0$, where $\omega_L$ is the laser frequency and $\omega_0$ is the 0th resonance of the resonator that the laser is pumping, changes the excitation energy ($E = \hbar\Delta$ in the rotating frame of the 0th resonance) of the pump signal (Fig. 2b, blue curve on the left side). This corresponds to the excitation of different modes of the band structure of the crystal (Fig. 2b, blue curve on the right side), leading to two synthetic Bloch waves with wave vectors $k_\pm$ given by:

$$k_\pm = \pm \frac{1}{a} \cos^{-1} \frac{\Delta}{\Omega} \qquad (1)$$

where $a = $ FSR is the lattice constant of the frequency crystal. For example, when $\Delta = 0$, two Bloch waves with wave vectors $k_\pm = \pm 0.5 \frac{\pi}{a}$ will be excited, representing waves that propagate along the positive and negative direction in frequency crystal (Fig. 3a) with a propagation

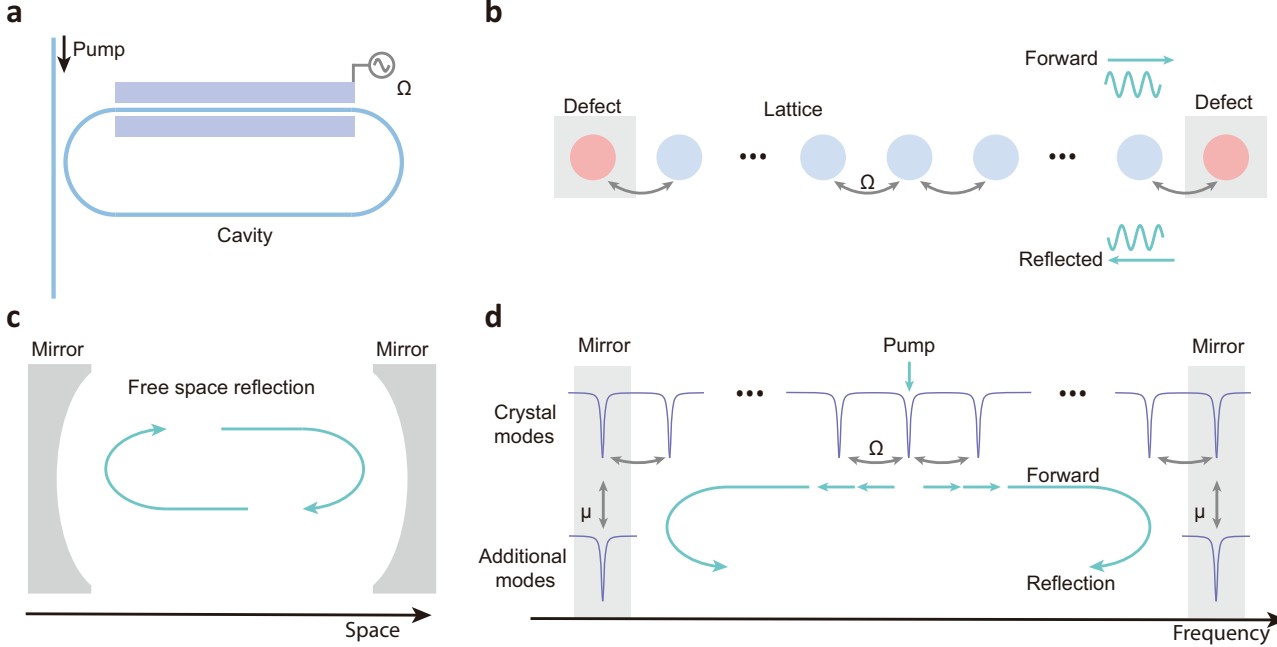

**Fig. 1 | Concept of the frequency mirror and mirror-induced reflection in a frequency crystal. a** Optical frequency crystals are generated using an optical resonator with electro-optic modulation. Microwave modulation, with frequency commensurate with the FSR of the resonator, creates nearest-neighbor coupling $\Omega$ between adjacent frequency modes of the resonator. **b** Defects introduced in the frequency crystal result in reflections of optical signal propagating in the synthetic

frequency dimension. **c** Reflection in the frequency domain is analogous to the spatial reflection. **d** Frequency mirror can be realized by coupling additional frequency modes to the crystal frequency modes. These additional modes will cause a mode-splitting effect, thus breaking the periodic translational symmetry of the frequency crystal, creating effective reflections of light in the frequency domain.

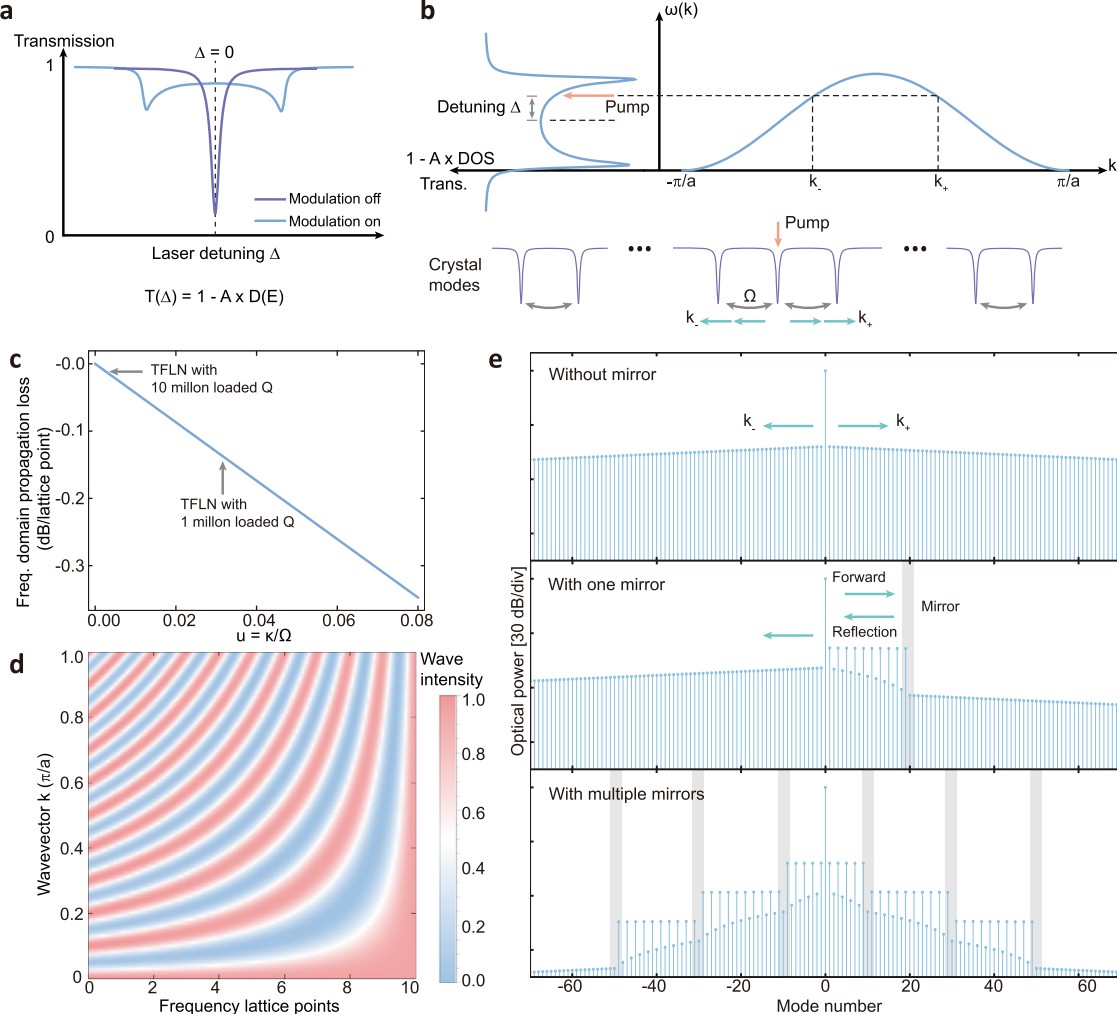

**Fig. 2 | Theory of the mirror-induced reflection and the trapped states in frequency domain. a** Illustration between the transmission spectrum of the resonator and density of states of the frequency crystal. When the modulation is off, the transmission shows a normal Lorentzian shape. When modulation is on, a frequency crystal is formed and each resonance is broadened. The transmission represents a direct link to the DOS, $T(\Delta) = 1 - A \times D(E)$, in which $T(\Delta)$ is the transmission, $D(E)$ is the DOS, and $A$ represents a factor for DOS (see detailed form in ref. 16). **b** With modulation on, the resonance shape corresponds to the density of states of the modes propagating in formed frequency crystal. Then, by controlling the laser detuning $\Delta$, the wave vectors of the Bloch waves can be controlled. In the general case, detuning launches two Bloch waves propagating along the positive and negative direction of the frequency crystal, with wavevectors $k_+$ and $k_-$, respectively, as determined by the band structure of the crystal. **c** Propagation loss

of the energy inside the frequency crystal as a function of parameter $u = \kappa/\Omega$. **d** Simulated energy distribution of the states that with a single frequency mirror. The reflected wave interferes with the forwarded wave, forming constructive/ destructive interference in the frequency domain. The period of the interference is determined by the wavevector. **e** When there is no mirror in the frequency crystal (top panel), the energy spreads through the EO modulation along the frequency dimension with a linear loss in dB scale. By applying one mirror, the Bloch waves are reflected, forming constructive/destructive interference (middle panel) with the optical energy located in every other lattice point enhanced/suppressed. When there are multiple mirrors, the light forms a trapped state in the frequency crystal (bottom panel), leading to flat spectrum for constructive interference in between mirrors.

phase of $\phi_p = k_\pm \times a = \pm 0.5\pi$ for a single hopping. To form a frequency mirror, additional mode $b$ is used to break the periodic translation symmetry. We assume mode $b$ (with a linewidth $\kappa_b$) is placed at frequency $\omega_{mr}$ that is frequency-degenerate with the crystal mode $a_{mr}$ (with a linewidth $\kappa$) and the coupling strength between $b$ and $a_{mr}$ is $\mu$. This additional mode $b$ plays the role of the mirror with a reflection coefficient:

$$r = -\frac{1 - \xi^2}{1 + \xi^2} \qquad (2)$$

where $\xi \approx -\frac{1}{1+(1+G)u}$. The parameter $G = 4\mu^2/\kappa_b\kappa$ is used to qualify the strength of the mirror ($G \sim 200 - 4300$ in this work. See Methods for details), and we assumed $u \equiv \frac{\kappa}{\Omega} \ll 1$ (see details in Methods). This leads to interference between the forward propagating and the reflected

waves (Fig. 1b) resulting in the final state:

$$\psi(x) \sim e^{ikx} + r\, e^{ikx_{mr}} e^{-ik(x-x_{mr})} \qquad (3)$$

where $k = k_\pm + i\frac{\alpha}{2}$, $x_{mr}$ represents the position of the mirror, and $\alpha$ is related to the propagation loss of the Bloch wave in the frequency domain. The propagation loss $L = e^{-\alpha a}$ is defined as the power loss for a single hop and determined by the coupling strength $\Omega$ and linewidth of the resonator $\kappa$:

$$L \approx \left| \left(1 - \frac{u}{2} - \frac{u^2}{8}\right) \right|^2 \qquad (4)$$

In our TFLN platform we estimate the propagation loss $L$ is 0.1dB per lattice point with $u = 0.024$ (Fig. 2c), which is low enough to

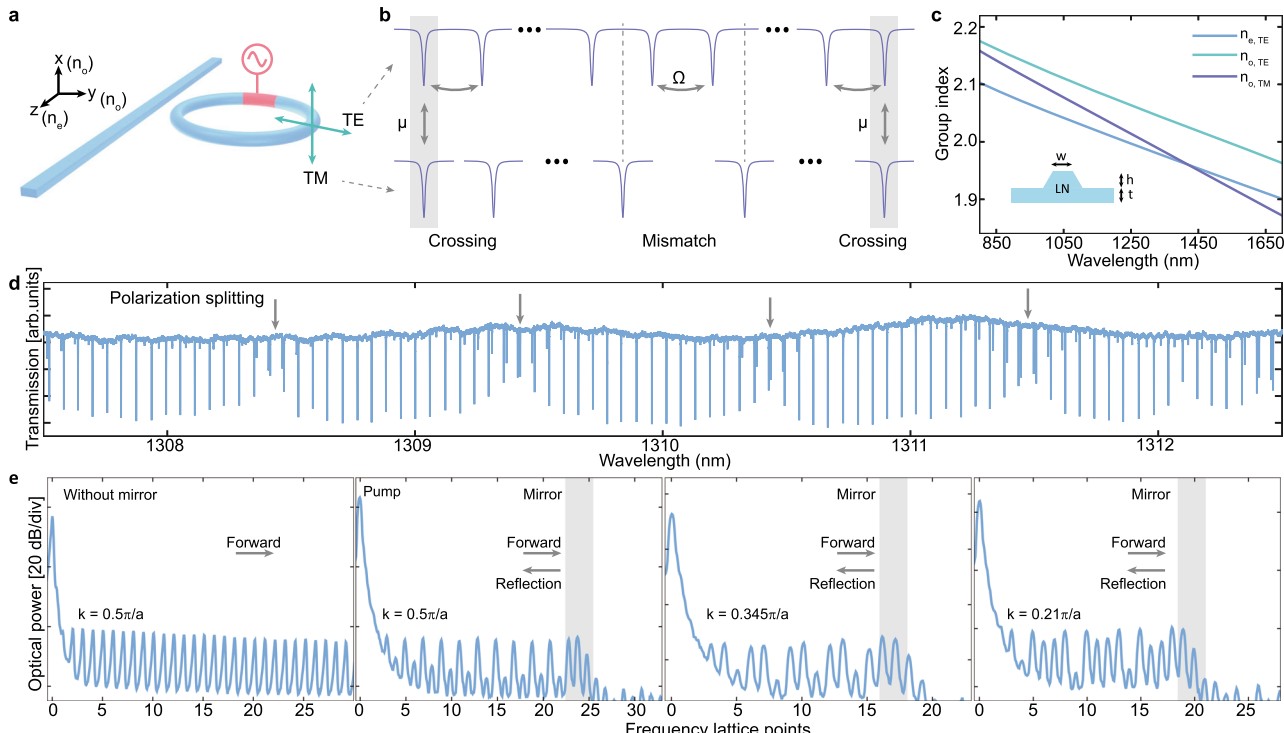

**Fig. 3 | Reflection induced interference in the frequency synthetic crystal using polarization coupling. a** On the x-cut thin-film lithium niobate, TE mode that propagates along the y- and z-direction has different indexes $n_{o,TE}$ and $n_{e,TE}$, while the TM mode has index of $n_{o,TM}$ for all directions. Dispersion engineering can be used to make the indices of TE and TM modes degenerate, leading to strong coupling $\mu$ between TE and TM resonances. **b** The frequency mirrors induced by mode-splitting only happens when the TE and TM modes have both index and frequency degeneracy. Difference in FSR of TE and TM modes, guarantees that they frequency overlaps at some frequencies, leading to formation of periodic frequency mirrors. **c** The group index $n_{o,TM}$ can be designed to be between the $n_{o,TE}$ and $n_{e,TE}$ via dispersion engineering. Then, TE mode circulating inside a x-cut lithium niobate resonator, it experiences different averaged indices (ranging from $n_{o,TE}$ to $n_{e,TE}$) at different bending points of the resonator. As a result, index degeneracy between TE and TM modes can be achieved over a broad wavelength range (850 nm to 1450 nm for $w = 1.4\mu m$, $h = 350$ nm, $t = 250$ nm). **d** Measured transmission spectrum of TE modes on the x-cut dispersion-engineered lithium niobate device. The mode-splitting breaks the translation symmetry of the crystal, leading to frequency mirrors. Arb. units: arbitrary units. **e** Experimental verification of the reflection due to frequency mirror using polarization-crossing. Optical energy propagates along the frequency dimension when there is no mirror. Applying frequency mirror leads to interference states and varying the Bloch wavevectors can adjust the shape of the state. Due to the discrete nature of the crystal, our output signal measures the oscillation with a discrete sampling in frequency domain with a sampling period equal to the lattice constant. As a result, for $k = 0.5\pi/a$, the energy distribution on each lattice point shows constructive/destructive interference at every other lattice points. The patterns for $k = 0.345\pi/a$ and $k = 0.21\pi/a$ correspond to different oscillation period compared to the case of $k = 0.5\pi/a$, leading to destructive interference at every three/four lattice points.

observe the interference and trapped state effects. With the above expression for the final state $\psi(x)$, we show such interference causes an oscillation of energy distribution $|\psi(x)|^2$ along the frequency dimension and the oscillating period is determined by the wave vector $k$ (Fig. 2d). Using the Heisenberg-Langevin equation, we numerically show that constructive/destructive interference in the frequency domain (see Methods) leads to trapped states using multiple mirrors (Fig. 2e). The mirror provides a sharp cut-off to the propagation and a 25.9dB power drop after passing the mirror. The mirror reflectivity is 0.994.

## Experiment

The first approach that we use to realize frequency domain mirrors is based on polarization mode coupling inside a dispersion-engineered TFLN micro-resonator. This requires both refractive index and frequency degeneracy of TE-like and TM-like modes (from here on referred to as TE and TM modes, respectively) propagating inside the ring (Fig. 3a, b). The group index degeneracy provides large $\mu$ while frequency degeneracy leads to mode splitting. Note that lithium niobate is a material with birefringence. As a result, for the optical modes that propagate along the y- and z-direction crystal axes of x-cut TFLN, the TE (y-propagation and z-propagation) modes have different indices $n_{o,TE}$ and $n_{e,TE}$, respectively. The TM (y-propagation and z-propagation) indices are both $n_{o,TM}$. Therefore, by optimizing the cross-

section of an x-cut lithium niobate micro-resonator, the value of $n_{o,TM}$ can be designed to be between the values of $n_{o,TE}$ and $n_{e,TE}$ over a broad range of wavelengths (Fig. 3c). The TE mode can have different average indices in the range of $n_{e,TE}$ to $n_{o,TE}$ at different spatial points within the bending region of the resonator. As a result, the TM modes can have an index degeneracy with the TE modes over a broad wavelength range (Fig. 3c). Frequency degeneracy is accomplished using a Vernier effect caused by the difference in FSR of TE and TM modes: the TM modes come in resonance with TE modes periodically, leading to periodic mode splitting that gives rise to periodic frequency mirrors (Fig. 3b). This coupling can be observed in the transmission spectrum of the TE modes (Fig. 3d).

To experimentally verify the presence and reflection of Bloch waves, we excite the frequency crystal at different values of detuning $\Delta$. By pumping at $\Delta = 0$ on the device without mirrors (no polarization induced splitting) the energy propagates along the frequency dimension (Fig. 3e) without a reflection. However, when the device with engineered polarization-splitting is used, propagating wave are reflected by polarization-splitting induced mirror, and interference between the two waves leads to a constructive/destructive pattern at every other lattice point due to the propagation phase of $\phi_p = \pm 0.5\pi$. Note that the constructive interference results in a flat spectrum of generated comb signal which could be of interest for frequency comb applications. By varying the laser detuning $\Delta$, we show varying

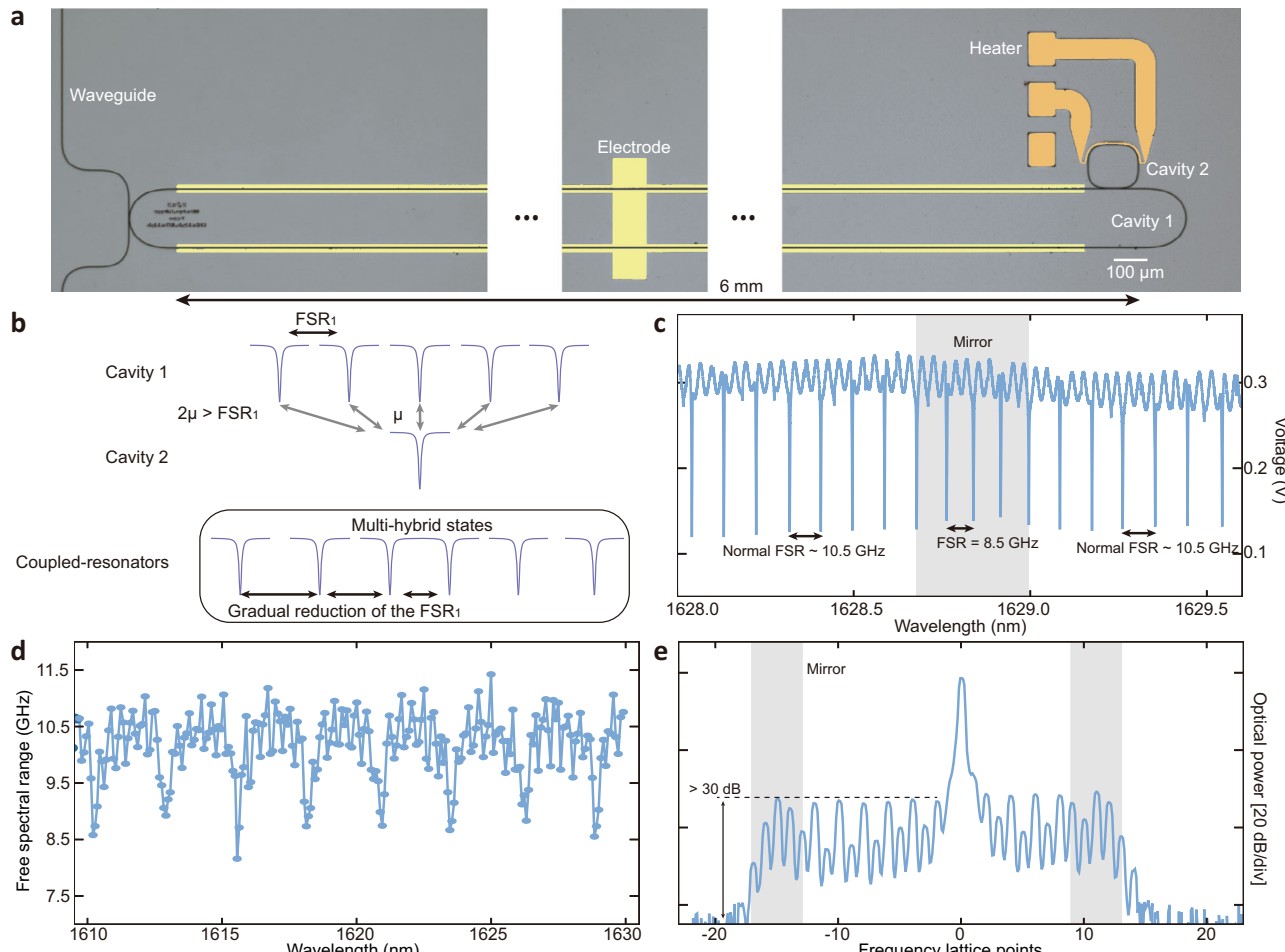

**Fig. 4 | Forming trapped states using frequency mirror implemented by coupled-resonators. a** Optical image (false-color) of the device. A long racetrack shape cavity (cavity 1) with $FSR_1$ of 10.46 GHz is used to generate the frequency crystal. A rectangular shape (cavity 2) with $FSR_2$ of 302.9 GHz is coupled to the cavity 1 to provide the frequency mirrors. Metal electrodes (light yellow) provides the efficient microwave modulation. Thermal heater (orange) can tune the position of the mirrors by varying the resonances of cavity 2. **b** Illustration of the coupling between two resonators when the coupling strength satisfies $2\mu > FSR_1$ or $\mu$ is comparable with $FSR_1$. In this case, rather than having two-mode splitting of resonances of cavity 1, the single resonance of cavity 2 couples to multiple resonances of cavity 1, forming multi-hybrid modes and leading to a gradual reduction of $FSR_1$. **c** Measured transmission spectrum of the device. The $FSR_1$ gradually changes from ~10.5 GHz (uncoupled value) to ~8.5 GHz and then goes back to ~10.5 GHz, indicating the strong coupling in presence of the resonance of cavity 2. **d** $FSR_1$ as a function of wavelength. The $FSR_1$ features dips with periodicity equal to the $FSR_2$, thus confirming the strong coupling between the two cavities. **e** Reflection at the mirror leads to constructive/destructive interference at every other lattice points. In addition, the mirror formed by coupling of multiple resonances supports trapped state, and provides >30 dB suppression of transmitted optical energy.

interference fringes, due to the change of wave vector $k$ (Fig. 3e). This polarization mirror shows a power cut-off of 16 dB (15.2 dB in simulation) with a reflectivity of 0.94. The reflectivity that the polarization crossing approach provided is limited since this coupling originates from fabrication imperfection-induced perturbation.

Even better control of defects in the synthetic frequency dimension can lead to realization of frequency mirrors with controllable reflection strength and position in the crystal as well as more complex arbitrary multi-mirrors configuration. Such control and strong mirror reflectivity can be achieved in TFLN using the coupled-resonator platform (Fig. 4a). In our design, a long racetrack cavity (cavity 1) with a $FSR_1 = 10.5$ GHz is used to generate the frequency crystal through electro-optic modulation, while a small square-shaped cavity (cavity 2) with a $FSR_2 = 302.9$ GHz is coupled to the racetrack cavity to provide frequency mirrors through the resultant mode splitting. Interestingly, in our system, the coupling strength between two cavities $\mu$ can be quite large, even comparable with the $FSR_1$, and as a result, a single resonance mode of the cavity 2 couples to multiple resonances of cavity 1 (Fig. 4b). This does not lead to a conventional two-mode-splitting but instead results in dispersive interactions that gradually

reduce $FSR_1$ in the frequency range around the resonances of cavity 2 (Fig. 4b). Indeed, the transmission spectrum of the device shows that the $FSR_1$ gradually varies from ~10.5 GHz to ~8.5 GHz and back to ~10.5 GHz at a wavelength around 1628.8 nm (Fig. 4c), corresponding to a 20% variation of the $FSR_1$. To verify that this large change of $FSR_1$ originates from the formation of multi-hybrid modes due to the presence of cavity 2, we measured the wavelength-dependence of $FSR_1$, and found that it is periodic with a period equal to $FSR_2$ (Fig. 4d). We extracted the coupling strength, finding it to be 6.8 GHz. This further verifies the existence of multi-hybrid modes since this system has $2\mu > FSR_1$ where $2\mu$ represents the conventional two-mode splitting. Such a strong coupling strength gives a mirror reflectivity of 0.999914. Finally, with the existence of multiple frequency mirrors, we verified the trapped state with constructive/destructive interference at every other lattice point in the coupled-resonator device (Fig. 4e). The strong mirror provides a cut-off of >30 dB for energy propagation in the frequency crystal (44 dB in simulation, measurement limited by noise floor). Despite the strong cut-off produced by the frequency mirrors, it is difficult to see multiple roundtrip effects within the two mirrors, due to the large propagation loss of our system (~0.15 dB/lattice point).

Constructive/destructive interference redistributes the trapped optical energy within the two mirrors, which could be useful for frequency-specific engineering of the frequency spectrum, while avoiding energy leakage to other frequencies. A list of all the relevant parameters of the polarization and coupled-resonator frequency mirrors are in Table S1.

## Discussion

Note that group velocity dispersion can lead to a gradually accumulated frequency detuning between the frequency comb line and the corresponding resonance frequency. Therefore, light that is spread over the frequency domain can eventually reach a "soft" frequency "boundary" when detuning becomes large. However, our platform[32] can generate several hundreds of lattice points without reaching this "soft" dispersion "boundary". Since this "soft boundary" does not result in a sharp discontinuity of the propagation, and introduces Bloch oscillations[33], it does not serve exactly as a frequency mirror. This effect has been previously observed in our TFLN system by applying a microwave detuning to effectively create large dispersion[16].

In summary, we have shown the reflection and trapped state of light in the frequency domain by introducing a mirror, that is, a defect, inside a frequency crystal. Our investigation utilizes the polarization mode-coupling to form mirrors in a single cavity and uses the coupled-resonator platform on TFLN[34] to achieve much stronger mirrors. We show that the reflection and trapped state can be formulated as the reflection of Bloch waves due to defect scattering and, can be tuned by varying the wavevectors of Bloch waves. Note that the TFLN platform features the lowest propagation loss in the frequency domain to date. The loss in this work (0.076 dB/lattice for the polarization system and 0.15 dB/lattice for the coupled-resonator system) is still higher, however, than the spatial propagation loss of light. Improving the quality factor of our TFLN rings from ~$10^6$ in this work to ~$10^7$ [35] with the same driving microwave power can reduce the propagation loss in the frequency domain, which is determined by $u = \kappa/\Omega$ ($\kappa$ is determined by quality factor and $\Omega$ is determined by microwave voltage), to 0.015 dB/lattice point, yielding a mirror reflectivity of >0.99999. Therefore, coupled resonators on TFLN can be promising to investigate multi-roundtrip dynamics in the frequency synthetic dimension, which may lead to the realization of a frequency domain cavity[36]. Introducing periodic mirrors via multiple additional resonators with lower propagation loss may lead to the realization a frequency domain photonic crystal[37,38]. Furthermore, the ability to control the distribution of light in the frequency synthetic dimension provides an advantageous way to manipulate the light frequency. For example, the trapped state in a synthetic frequency crystal can be used to generate flat-slope EO combs with better energy confinement in frequency domain, which is important for applications in spectroscopy, astronomy (astro-comb), and quantum frequency combs[39–43]. Finally, realizing frequency domain scattering beyond reflection and transmission, using a high-dimensional frequency crystal[1,16,19,23] or other crystal structures[18,20], could pave ways to investigate high-dimensional geometrical phases and topologies. Specifically, with the ability to introduce defects in the frequency domain, the recently emerged topological frequency comb[44] might can be combined with our coupled-resonator platform. TFLN can generate frequency combs using both EO and Kerr non-linearity simultaneously[32], leading to the realization of a spatial-frequency topological frequency comb. Our approach could form a basis for controlling the crystal lattice structure, band structure, and energy distributions in frequency domain.

## Methods
### Device fabrication
The devices are fabricated on a x-cut lithium niobate wafer (NANOLN). The wafer contains a 600 nm LN layer, 2 µm buried oxide, and a 500 µm Si handle. The optical layer is patterned using electron-beam lithography and the pattern is transferred to the lithium niobate layer using Ar⁺-based reactive ion etching with an etch depth of 350 nm. Metal electrodes are defined using optical lithography and deposited with electron-beam evaporation. The electrodes consist of 15 nm of Ti and 300 nm of Au. The oxide cladding is deposited using plasma-enhanced chemical vapor deposition. The heater layer (15 nm of Ti and 200 nm Pt) is also patterned by photolithography followed by electron-beam evaporation and bi-layer lift-off.

### Measurement
Telecommunication-wavelength light from a fiber-coupled tunable laser passes through a polarization controller and is coupled to the LN chip using a lensed fiber. The output is collected using an aspheric lens and an optical spectrum analyzer is used to characterize the output frequency spectrum. The microwave signal is generated and amplified before sending it to the electrode of the device using an electrical probe.

### Theory and simulation of the frequency-mirror-induced reflection and the trapped states
**Normal crystal dynamics without mirrors.** We first consider the crystal dynamics without frequency mirrors. The optical frequency crystal generated using electro-optic modulation on a single cavity can be described by its Hamiltonian in the following form:

$$H = \sum_{j=-N}^{N} \omega_j a_j^\dagger a_j + \Omega \cos\omega_m t \left( a_j^\dagger a_{j+1} + h.c. \right) \tag{5}$$

where $\omega_j$ is the frequency of each mode, $\Omega$ is the coupling rate induced by the microwave modulation, and $\omega_m$ is the frequency of the microwave modulation. The total number of frequency modes that are coupled are labeled by number $-N$ to $N$. Using the Heisenberg-Langevin equation, we derived the equations of motion of each mode $a_j$ in the frequency crystal:

$$\dot{a}_j = \left(-i\omega_j - \frac{\kappa}{2}\right)a_j - i\Omega\cos\omega_m t\left(a_{j+1} + a_{j-1}\right) - \sqrt{\kappa_e}\alpha_{in}e^{-i\omega_L t}\delta_{j,0} \tag{6}$$

where the $\kappa_e$ is the coupling rate between the waveguide and cavity, $\kappa$ is the total loss rage of $a_j$, $\alpha_{in}$ is the pump power, $\omega_L$ is the laser frequency, and $\delta_{j,0}$ is used to denote the pumped mode. The frequency of each mode $a_j$ is $\omega_j = \omega_0 + j \times$ FSR with $\omega_0$ representing the 0th mode. Such a set of equations can be solved by changing the rotating frame for each mode $a_j \to a_j e^{-i\omega_L t} e^{-ij\omega_m t}$ and doing the Fourier transformation to solve this equation in frequency domain:

$$0 = \left(i\Delta + i\delta - \frac{\kappa}{2}\right)a_j - i\frac{\Omega}{2}\left(a_{j+1} + a_{j-1}\right) - \sqrt{\kappa_e}\alpha_{in}\delta_{j,0} \tag{7}$$

where $\delta = \omega_m -$ FSR and $\Delta = \omega_L - \omega_0$ are the microwave detuning and laser detuning, respectively.

In the case of $\Delta = \delta = 0$, the equation for each mode $a_j$ is:

$$0 = -\frac{\kappa}{2}a_j - i\frac{\Omega}{2}a_{j+1} - i\frac{\Omega}{2}a_{j-1} - \sqrt{\kappa_e}\alpha_{in}\delta_{j,0} \tag{8}$$

Note that the equation for the frequency mode $a_N$ with a mode number of $N$ is:

$$0 = \left(-\frac{\kappa}{2}\right)a_N - i\frac{\Omega}{2}a_{N-1} \tag{9}$$

which leads to $a_N = -i\frac{\Omega}{\kappa}a_{N-1}$. As a result, the equation for $a_{N-1}$ is:

$$0 = \left(-\frac{\kappa}{2}\right)a_{N-1} - i\frac{\Omega}{2}a_{N-2} - \frac{\Omega}{2}\frac{\Omega}{\kappa}a_{N-1} \tag{10}$$

Therefore, we obtained another relation $a_{N-1} = -i\frac{\Omega}{\kappa(1+\frac{\Omega^2}{\kappa^2})}a_{N-2}$. The relation for an arbitrary mode can be obtained using iteration:

$$a_l = -i\frac{\Omega}{\kappa}\frac{1}{1+\frac{\frac{\Omega^2}{\kappa^2}}{1+\frac{\frac{\Omega^2}{\kappa^2}}{1+\frac{\Omega^2}{\kappa^2}}}}a_{l-1} \tag{11}$$

where we have the total number of the factor $\frac{\Omega^2}{\kappa^2}$ as $N-l$. Using this relation for an arbitrary mode, the equation of motion for the pump mode (0th mode) can be obtained as:

$$0 = -\frac{\kappa}{2}a_0 - i\frac{\Omega}{2}\left(-i\frac{\Omega}{\kappa}\frac{1}{1+\frac{\frac{\Omega^2}{\kappa^2}}{1+\frac{\frac{\Omega^2}{\kappa^2}}{1+\frac{\Omega^2}{\kappa^2}}}}\right)a_0 - i\frac{\Omega}{2}\left(-i\frac{\Omega}{\kappa}\frac{1}{1+\frac{\frac{\Omega^2}{\kappa^2}}{1+\frac{\frac{\Omega^2}{\kappa^2}}{1+\frac{\Omega^2}{\kappa^2}}}}\right)a_0 - \sqrt{\kappa_e}\alpha_{in} \tag{12}$$

This equation is equivalent to:

$$0 = \left(-\frac{\kappa}{2} - \frac{\kappa_{MW}}{2}\right)a_0 - \sqrt{\kappa_{e2}}\alpha_{in} \tag{13}$$

in which $\kappa_{MW} = \kappa \times 2(f_n - 1)$ with:

$$f_n = 1 + \frac{\frac{\Omega^2}{\kappa^2}}{1+\frac{\frac{\Omega^2}{\kappa^2}}{1+\frac{\frac{\Omega^2}{\kappa^2}}{1+\frac{\Omega^2}{\kappa^2}}}} \tag{14}$$

When $N$ is very large, the limit of $f_n$ can be used: $\lim_{n\to\infty} f_n = f$. As a result, we have $f = 1 + \frac{\frac{\Omega^2}{\kappa^2}}{f}$, which can be used to solve the final expression for $f$ and $\kappa_{MW}$:

$$f = 1 + \frac{\frac{\Omega^2}{\kappa^2}}{1+\frac{\frac{\Omega^2}{\kappa^2}}{1+\frac{\frac{\Omega^2}{\kappa^2}}{1+\frac{\Omega^2}{\kappa^2}}}} = \frac{1+\sqrt{1+\frac{4\Omega^2}{\kappa^2}}}{2} \tag{15}$$

$$\kappa_{MW} = \kappa\left(\sqrt{1+\frac{4\Omega^2}{\kappa^2}} - 1\right) \tag{16}$$

The $\kappa_{MW}$ is the effective loss rate for the pump mode that generated by the microwave modulation[32]. With the expression of $f$, we can simplify the relation between $a_l$ and $a_{l-1}$ as:

$$a_l = -i\frac{1}{uf}a_{l-1} \tag{17}$$

where $u \equiv \kappa/\Omega$ and $f = \frac{1+\sqrt{1+4/u^2}}{2}$.

This gives the propagation loss:

$$L = e^{-\alpha a} \equiv \left|\frac{a_l}{a_{l-1}}\right|^2 = \left|\frac{1}{uf}\right|^2 = \frac{2}{u+2\sqrt{1+\frac{u^2}{4}}} \approx 1 - \frac{u}{2} - \frac{u^2}{8} \tag{18}$$

**Frequency crystal with frequency mirrors.** The frequency mirror can be introduced by coupling additional modes $b_k$ ($k=1,2,\dots$) to the frequency crystal. To derive the reflection and transmission coefficient

for the mirror, we consider a problem that a single additional mode $b$ is coupled to a frequency mode $a_{mr}$ with a coupling strength $\mu$ and $b$ is frequency-degenerate with $a_{mr}$. Further, we consider the case that the pump source is far away from the mirror in frequency domain, i.e., the pump frequency is far away from the frequency of $b$. The equation of motions for $a_{mr}$ and $b$ are:

$$0 = \left(-\frac{\kappa}{2}\right)a_{mr} - i\frac{\Omega}{2}\left(a_{mr+1} + a_{mr-1}\right) - i\mu b \tag{19}$$

$$0 = \left(-\frac{\kappa_b}{2}\right)b - i\mu a_{mr} \tag{20}$$

Note that $b$ is in a rotating frame of $a_{10}$ since they are frequency degenerate. The steady-state solution of the above equations for each frequency mode and additional modes gives the simulated energy distribution of frequency crystals with mirrors.

The mirror mode splits frequency space into two different regions: region that contains both input and reflected waves and the region that contains only the transmitted wave. Therefore, the region of frequency space that contains the transmitted wave should follow the normal dynamics of the frequency crystal, i.e., free propagation without reflection. As a result, we conclude that $a_{mr}$ and $a_{mr+1}$ will obey the relation:

$$a_{mr+1} = -i\frac{1}{uf}a_{mr} \tag{21}$$

At the same time, we have the following equations:

$$0 = -\frac{\kappa}{2}a_{mr} - i\frac{\Omega}{2}a_{mr+1} - i\frac{\Omega}{2}a_{mr-1} - i\mu b \tag{22}$$

$$0 = \left(-\frac{\kappa_b}{2}\right)b - i\mu a_{mr} \tag{23}$$

With the Eqs. (21)–(23) we find:

$$\frac{a_{mr+1}}{a_{mr-1}} = -\frac{1}{u^2 f(1+G)+1} \tag{24}$$

where $u \equiv \kappa/\Omega$, $f = \frac{1+\sqrt{1+4/u^2}}{2}$, and $G = 4\mu^2/\kappa_b\kappa$ is the parameter used to qualify the strength of the mirror.

For the region that contains both the input and reflected waves, due to the interference, we have:

$$a_{mr-1} = (1+r)A_0 \tag{25}$$

where $A_0$ represents the input wave amplitude at the position of mirror.

The transmission and reflection coefficients are:

$$t = \frac{a_{mr+1}}{A_0} \tag{26}$$

$$r^2 = 1 - t^2 \tag{27}$$

Using Eqs. (24)–(27), the reflection coefficient is:

$$r = -\frac{1-\xi^2}{1+\xi^2} \tag{28}$$

with $\xi = -\frac{1}{u^2 f(1+G)+1}$. In the regime that $u \ll 1$, we have $\xi \approx -\frac{1}{\left(u+\frac{u^2}{2}+\frac{u^3}{8}\right)(1+G)+1} \approx -\frac{1}{1+(1+G)u}$.

Finally, we have:

$$R = |r|^2 = \frac{\left(1 - \xi^2\right)^2}{\left(1 + \xi^2\right)^2} \tag{29}$$

$$T = |t|^2 = \frac{4\xi^2}{\left(1 + \xi^2\right)^2} \tag{30}$$

To simulate the frequency crystal with arbitrary additional mirrors, i.e., several additional modes $b_k$ ($k = 1, 2, \ldots$) coupled to the crystal, we use the equation:

$$\dot{a}_{mr,k} = \left(i\Delta + i\delta - \frac{\kappa}{2}\right)a_{mr,k} - i\frac{\Omega}{2}\left(a_{mr,k+1} + a_{mr,k-1}\right) - i\mu b_k \tag{31}$$

$$\dot{b}_k = \left(i\Delta + i\delta - \frac{\kappa}{2}\right)b_k - i\mu a_{mr,k} \tag{32}$$

where $a_{mr,k}$ is the mode that is frequency degenerate with $b_k$. We combine the above equations with the equations for modes that are not coupled to the mirrors:

$$\dot{a}_j = \left(i\Delta + i\delta - \frac{\kappa}{2}\right)a_j - i\frac{\Omega}{2}\left(a_{j+1} + a_{j-1}\right) - \sqrt{\kappa_e}\alpha_{in}\delta_{j,0} \text{ for } j \neq k, \tag{33}$$

to numerically simulate the frequency crystals with mirrors.

**Note added to proof**

In the process of writing this manuscript another group reported the observation of frequency boundaries in a fiber-cavity system[45].

## Data availability

The datasets generated and analyzed during the current study are available from the corresponding authors on reasonable request.

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

## Acknowledgements

This work is supported by ARO W911NF2010248 (Y.H.), NSF QuIC TAQS OMA-2137723 (Y.H.), DARPA LUMOS HR0011-20-C-0137 (M.Y., R.C., and M.L.), AFRL Quantum Accelerator FA9550-21-1-0056 (N.S.), ONR N00014-22-C-1041 (M.Y. and R.C.), NASA 80NSSC21C0583 (M.Y. and R.C.), NIH 5R21EY031895-02 (M.L.), Harvard Quantum Initiative (D.Z.), Research Grants Council, University Grants Committee (CityU 11212721) (C.W.). N.S. acknowledges support from the AQT Intelligent Quantum Networks and Technologies (INQNET) research program.

## Author contributions

Y.H. conceived the idea, developed the theory, performed the simulation of frequency mirrors. M.Y. performed the dispersion simulations and carried out the measurement of the polarization mirrors. M.Y. and Y.H. measured the coupled-resonator mirrors. Y.H. and R.C. fabricated the device. Y.H. wrote the manuscript with contributions from all authors. N.S., D.Z., and C.W. helped with the project. M.L. supervised the project.

## Competing interests

M.L. are involved in developing lithium niobate technologies at Hyper-Light Corporation. The remaining authors declare no competing interests.
