## [Peer review file · Nature Communications]

Reviewer #1 (Remarks to the Author):

This demonstrates the mirror behaviors in a frequency synthetic dimension lattice formed in a TFLN ring cavity. Although the on-chip synthetic dimension has been demonstrated in some prior studies, the TFLN ring incorporating a phase modulator is a powerful tool that achieves a very low transition loss between lattices. Moreover, the idea shown in this paper for the mirror formation is impressive not only as a demonstration of synthetic mirror but as a practical technique that makes a flat spectrum, desired bandwidth, high-efficiency frequency comb.

The paper is simply written, easy to read, and the figures are well understood. One small question is that the coupling strength between the TE and TM. The flexible coupling between the TM mode and TE mode whose effective index changes repeatedly in the ring is a nice idea, but the coupling may occur in a limited part of the ring and the coupling cannot be so high. I would ask the authors to give some comment on this matter.

Fundamentally, I recommend the acceptance of this paper for publication.

Reviewer #2 (Remarks to the Author):

Comments to the Authors

In the manuscript entitled "Mirror-induced reflection in the frequency domain", Yaowen Hu and co-authors investigate theoretically and experimentally the interesting area of synthetic dimension by adjusting light guidance characteristics and adjusting frequency mode conversion in a thin-film lithium niobate (TFLN) micro-resonator.

In this paper, the authors developed the model for frequency reflection (i.e. frequency domain mirror) relying on Bloch waves theory in case of the addition of an "exotic" mode to break periodic translation symmetry. This allows the trapping of a selected frequency range in a particular system (herein, micro-resonator). Interestingly, from an experimental point of view, the authors leverage the polarization properties of the TFLN, which can be readily controlled by using the heater and coupled resonant cavities, to efficiently tailor the properties of the frequency mirrors. The results herein are significant from a fundamental viewpoint with strong interests in e.g. topological photonics and this experimental demonstration underline numerous potential applications for e.g. tunable frequency combs and metrology.

Overall, the manuscript is well written with clear information that will be of interest to the readers. Therefore, I recommend this manuscript for publication in Nature Communications. Below, I append a few minor comments that the authors may consider before publication.

Minor comments:

1. Could the authors why one can see 20 dB power drop after passing the mirror in Fig.2 and 3, while, in Figure 4, the authors claimed "...30 dB suppression of transmitted optical energy"? Can the authors explain these discrepancies and whether there is agreement with numerical simulations in this case?
2. The main limitation of the used TFLN is the high loss of hopping between the lattice points. Consequently, the author claimed that "it is difficult to see multiple roundtrip effects within the two mirrors". To improve this, the authors propose to use the TLFN with high factor quality and improve the microwave driving power. Improving the ring factor can improve the linewidth of the crystal modes (κ). However, it is not clear to me how microwave driving power changes the coupling strength between nearest-neighbor lattice points (Ω) and/or κ . Could the authors maybe comment further on that?
3. The authors built a model for the frequency mirrors by assuming a large number of modes (N). Could the authors roughly estimate/compare the properties of the mirrors (e.g., reflection coefficient, the position of mirrors in frequency domain) if the boundary condition shown in ref. [42] is used?

Reviewer #3 (Remarks to the Author):

Overall, a timely and interesting work. The authors report a synthetic crystal in which a defect is introduced to construct light-reflecting boundaries. They take advantage of the introduction of coupled ring resonators and electro-optical modulation to demonstrate the mirror formation at the boundaries in this frequency crystal and realize trapped states between adjacent synthetic mirrors. To ensure their system holds the promise they claim in terms of scalability and application, however, the report can benefit from further clarification and quantitative evaluation of loss and efficiency. Another aspect of the manuscript that requires further clarification is the boundary type they have designed and observed, especially in comparison to the recent related results reported elsewhere.

Detailed questions/comments are as follows:

- 1) Calling the parameter C defined in the manuscript might be misleading. The author mention that this parameter is "analogous to the cooperativity in cavity quantum electrodynamics". I disagree with this statement. The cooperativity in the cavity-QEC characterizes the strength of a "two-level system" to a

cavity mode and not a simple harmonic oscillator. For example, for large C one expects non-classical effects (e.g. anti-bunching of photons) to occur. While in the presented material everything can be explained by nonlinear optical equations, with no quantum effect, as the authors have correctly done in the manuscript. My only comment is on the usage of the term cooperativity.

2) Naively, one expects a frequency walk-off across different modes due to group velocity dispersion. This makes the “cold” frequency resonances non-equidistant. It would be helpful to have a discussion on this.

3) The authors make a note in the paragraph starting from line 158 that the coupling strength between the two cavities presented in Fig. 4 can be stronger than the FSR of the cavity¹, leading to the presented non-conventional gradual reduction of the FSR of that cavity. They may quantitatively further discuss the relationship between the degree of coupling and observed gradual reduction. They may also wish to provide further supplementary discussion on cases where the coupling strength is weaker or equal to the FSR¹.

4) Further upon the previous comment, they may wish to add general quantitative characterization parameters of their system, including waveguide-ring coupling efficiency and associate losses.

5) In the final paragraph of the manuscript, the authors argue the potential utilization of their system for higher-order topological physics. Evaluating with the recent relevant results reported in the literature (<https://doi.org/10.48550/arXiv.2203.11296>), in which the constructed synthetic boundaries are comparatively sharp, they might consider enriching their discussion of such potential applications so that it more precisely matches their specific type of reported frequency boundaries. A more relevant direction might be the investigation of topological frequency combs (<https://www.nature.com/articles/s41567-021-01302-3>) in such synthetic systems.

6)The authors make the claim that their system may be useful for the realization of higher frequency crystals. However, with the significant observed propagation loss, evaluation of the scalability of their system requires more investigation. They may wish to for instance estimate the required microwave driving power to achieve more round trips and multi-trapped states across the frequency crystal.

7)The sentence starting at line 129 seems to be grammatically incorrect.

Response to Reviewers' Comments:

We thank all the reviewers for spending their time assessing our work and providing valuable feedback to help us improve the manuscript. We have prepared a detailed point-by-point response to all the reviewers' comments (please see below).

Reviewer #1 (Remarks to the Author):

This demonstrates the mirror behaviors in a frequency synthetic dimension lattice formed in a TFLN ring cavity. Although the on-chip synthetic dimension has been demonstrated in some prior studies, the TFLN ring incorporating a phase modulator is a powerful tool that achieves a very low transition loss between lattices. Moreover, the idea shown in this paper for the mirror formation is impressive not only as a demonstration of synthetic mirror but as a practical technique that makes a flat spectrum, desired bandwidth, high-efficiency frequency comb.

Our response: We thank the reviewer for very positive assessment on our work and we agree with the reviewer that the mirror formation is important for not only synthetic mirrors but also practical applications.

The paper is simply written, easy to read, and the figures are well understood. One small question is that the coupling strength between the TE and TM. The flexible coupling between the TM mode and TE mode whose effective index changes repeatedly in the ring is a nice idea, but the coupling may occur in a limited part of the ring and the coupling cannot be so high. I would ask the authors to give some comment on this matter.

Our response: We thank the reviewer for helping us improve the manuscript. Indeed, the coupling only happens in the physical region of rings where the effective index of TE and TM modes are close, and since the coupling strength is actually determined by fabrication imperfections it might not be very strong. However, our lithium niobate cavity has high Q (loaded Q = 2.2 million) therefore this coupling strength is large enough to observe frequency boundary effects. For example, this coupling between TE and TM modes is 0.86 GHz at 1300 nm in our system, which is much larger than our resonance linewidth of 105 MHz at 1300 nm.

The low coupling strength of polarization crossing can be addressed by the coupled-resonator approach. In our coupled-resonator device, we achieved a much stronger coupling (6.8 GHz), which leads to a strong reflectivity of the mirror (0.999914), compared to the reflectivity of 0.94 in the case of polarization crossing.

To address this point, we have added several sentences in our manuscript (see below in red):

In the paragraph of the results of polarization crossing: “This polarization mirror shows a power cut-off of 16 dB (15.2 dB in simulation) with a reflectivity of 0.94. The reflectivity that the polarization crossing approach provided is limited since this

coupling originates from fabrication imperfection-induced perturbation. A much stronger mirror can be realized by the coupled-resonator platform on TFLN (see below).”

In the paragraph of the results of coupled-resonators: “We extracted the coupling strength, and found it to be 6.8 GHz. This further verifies the existence of multi-hybrid modes since this system has $2\mu > \text{FSR}_1$ where 2μ represents the conventional two-mode splitting. Such a strong coupling strength gives a mirror reflectivity of 0.999914.”

Fundamentally, I recommend the acceptance of this paper for publication.

Our response: We thank the reviewer for recommendation of publication.

Reviewer #2 (Remarks to the Author):

Comments to the Authors

In the manuscript entitled “Mirror-induced reflection in the frequency domain”, Yaowen Hu and co-authors investigate theoretically and experimentally the interesting area of synthetic dimension by adjusting light guidance characteristics and adjusting frequency mode conversion in a thin-film lithium niobate (TFLN) micro-resonator.

In this paper, the authors developed the model for frequency reflection (i.e. frequency domain mirror) relying on Bloch waves theory in case of the addition of an “exotic” mode to break periodic translation symmetry. This allows the trapping of a selected frequency range in a particular system (herein, micro-resonator). Interestingly, from an experimental point of view, the authors leverage the polarization properties of the TFLN, which can be readily controlled by using the heater and coupled resonant cavities, to efficiently tailor the properties of the frequency mirrors. The results herein are significant from a fundamental viewpoint with strong interests in e.g. topological photonics and this experimental demonstration underline numerous potential applications for e.g. tunable frequency combs and metrology.

Our response: We thank the reviewer for the very positive assessment in our work and the detailed discussions on the impacts on both fundamental science and practical applications.

Overall, the manuscript is well written with clear information that will be of interest to the readers. Therefore, I recommend this manuscript for publication in Nature Communications. Below, I append a few minor comments that the authors may consider before publication.

Our response: We thank the reviewer for recommendation of publication. We have addressed the reviewer’s comments (see below) and we appreciate the reviewer helping us improve the manuscript.

Minor comments:

1. Could the authors why one can see 20 dB power drop after passing the mirror in Fig.2 and 3, while, in Figure 4, the authors claimed “...30 dB suppression of transmitted optical energy”? Can the authors explain these discrepancies and whether there is

agreement with numerical simulations in this case?

Our response: We thank the reviewer for bringing up this confusion. The reason that polarization crossing has a 20 dB power cut-off and Fig. 4 has 30 dB power cut-off is due to the difference in the mirror strength between the polarization mirror and coupled-resonator mirrors.

For example, our polarization crossing approach has a coupling strength μ of 0.86 GHz, and parameter $G = 268$, and mirror reflectivity of 0.94, while our coupled resonator platform has a coupling strength $\mu = 6.8$ GHz, $G = 4296$, mirror reflectivity of 0.999914. The difference between the achieved coupling strength and the quality factor of the devices leads to this difference in power cut-off.

To address this confusion, we have listed all of the system parameters in a table in supplementary (see below in red):

Table S1 | Parameters of the systems

Parameters	Simulation in Fig. 2	Polarization mirror in Fig. 3	Coupled-resonator mirror in Fig. 4	Outlook
Waveguide-ring coupling κ_e (MHz)	48	20	39	2
Intrinsic loss rate κ_i (MHz)	96	85	169	19
Resonance linewidth κ (MHz)	144	105	207	21
Intrinsic Q	2.0×10^6	2.7×10^6	1.1×10^6	10×10^6
Loaded Q	1.3×10^6	2.2×10^6	0.89×10^6	9.2×10^6
Coupling strength Ω (GHz)	6	6	6	6
Parameter $u = \kappa/\Omega$	2.4×10^{-2}	1.8×10^{-2}	3.5×10^{-2}	0.35×10^{-2}
Propagation loss L (dB/lattice)	0.1	0.076	0.15	0.015
Pump wavelength (nm)	1550	1309	1628	1550
Mirror coupling strength μ (GHz)	2	0.86	6.8	5
Free spectral range FSR (GHz)	10	10.5	10.5	10
Parameter G	772	268	4296	250627
Reflectivity	0.994	0.94	0.999914	0.999997
Power cut off, simulation (dB)	25.9	15.2	44	58.9
Power cut off, experiment (dB)	N/A	16	>30 (limited by noise floor)	N/A

2. The main limitation of the used TFLN is the high loss of hopping between the lattice points. Consequently, the author claimed that “it is difficult to see multiple roundtrip effects within the two mirrors”. To improve this, the authors propose to use the TFLN with high factor quality and improve the microwave driving power. Improving the ring factor can improve the linewidth of the crystal modes (κ). However, it is not clear to me how microwave driving power changes the coupling strength between nearest-neighbor lattice points (Ω) and/or κ . Could the authors maybe comment further on that?

Our response: We thank the reviewer for helping us improve the manuscript. We have added corresponding discussions in our manuscript to clarify the relationship between microwave driving power, Ω , κ , and Q . Please see below in red:

The relationship between Ω and applied microwave signal:

“The coupling strength between nearest neighbor lattice points, Ω , (Fig. 1b) is proportional to the voltage of the microwave driving signal and is related to the conventional modulation index β of a phase modulator as $\Omega = \frac{\beta}{2\pi} \text{FSR}$ (in a conventional modulator, the relationship between β and the driving voltage V is $\beta = \pi \frac{V}{V_\pi}$, in which V_π is the voltage required to achieve a π phase shift)”

How the improvement of TFLN ring quality helps the propagation loss and reflectivity:

“Note that the TFLN platform features the lowest propagation loss in the frequency domain to date. The loss in this work (0.076 dB/lattice for the polarization system and 0.15 dB/lattice for the coupled-resonator system) is still higher, however, than the spatial propagation loss of light. Improving the quality factor of our TFLN rings from $\sim 10^6$ in this work to $\sim 10^7$ [35] with the same driving microwave power can reduce the propagation loss in the frequency domain, which is determined by $u = \kappa/\Omega$ (κ is determined by quality factor and Ω is determined by microwave voltage), to 0.015 dB/lattice point, yielding a mirror reflectivity of >0.99999 . Therefore, coupled resonators on TFLN can be promising to investigate multi-roundtrip dynamics in the frequency synthetic dimension, which may lead to the realization of a frequency domain cavity [36].”

Finally, this issue of high loss when hopping along the lattice is not specifically limited to TFLN but to all of the current frequency synthetic dimension platforms. In fact, TFLN holds the lowest propagation loss that has ever been achieved in the frequency synthetic dimension. This propagation loss is closely related to the EO frequency comb slope and comb span. For example, we achieved a slope of 0.7 dB/nm and span of 132 nm in our coupled-resonator EO comb [Hu et al. arXiv: 2111.14743 (2021)], and both the slope as well as span are largest over all the other existing EO combs. The demonstrated comb contains >500 frequency modes, which is large compared to other EO combs, such as <100 frequency modes in fiber system [Dutt et al. arXiv: 2203.11296]. Therefore, we believe, coupled-resonator on TFLN is potentially the most promising platform to achieve multi-roundtrip phenomenon in the future, provided an improvement in quality factors is obtained.

3. The authors built a model for the frequency mirrors by assuming a large number of modes (N). Could the authors roughly estimate/compare the properties of the mirrors (e.g., reflection coefficient, the position of mirrors in frequency domain) if the boundary condition shown in ref. [42] is used?

Our response: We thank the reviewer for raising this point. We have estimated the properties of mirrors in the reference [42] that used a fiber system. In their system, we estimate the linewidth of the resonance to be $\kappa = 0.1277$ MHz, coupling strength Ω due to microwave is $\Omega = 0.642$ MHz, coupling from mirrors $\mu = 0.426$ MHz, parameter $u = 0.2$, propagation loss in frequency domain $L = 0.86$ dB/lattice point, parameter $G = 57$, reflectivity = 0.99 and a power cut-off on mirror = 22.7 dB.

Comparing to the fiber system, our system has a much smaller $u \sim 0.01$ and propagation loss $L \sim 0.1$ dB/lattice. We can achieve a much stronger mirror reflectivity 0.999914 using coupled-resonator system while preserving the low propagation loss. Both the fiber system and our system can be tuned, therefore there is no significant difference of the position of mirrors. However, our system is on an integrated platform, and therefore could be promising to scale to multi-resonator systems.

Reviewer #3 (Remarks to the Author):

Overall, a timely and interesting work. The authors report a synthetic crystal in which a defect is introduced to construct light-reflecting boundaries. They take advantage of the introduction of coupled ring resonators and electro-optical modulation to demonstrate the mirror formation at the boundaries in this frequency crystal and realize trapped states between adjacent synthetic mirrors. To ensure their system holds the promise they claim in terms of scalability and application, however, the report can benefit from further clarification and quantitative evaluation of loss and efficiency. Another aspect of the manuscript that requires further clarification is the boundary type they have designed and observed, especially in comparison to the recent related results reported elsewhere.

Our response: We thank the reviewer for the positive assessment of our work. We agree with the reviewer that further quantitative evaluation and clarification of the system as well as comparing with recent related results can help improve our manuscript. We therefore have made several substantial changes to our manuscript, including a table that contains all the detailed parameters of the system, as well as an outlook. With these added materials, we believe the manuscript has been improved and the advantage of our system is more clear. We hope the reviewer feels the same too. Please see our point-by-point response below.

Detailed questions/comments are as follows:

1) Calling the parameter C defined in the manuscript might be misleading. The author mention that this parameter is “analogous to the cooperativity in cavity quantum electrodynamics”. I disagree with this statement. The cooperativity in the cavity-QEC characterizes the strength of a “two-level system” to a cavity mode and not a simple harmonic oscillator. For example, for large C one expects non-classical effects (e.g. anti-bunching of photons) to occur. While in the presented material everything can be explained by nonlinear optical equations, with no quantum effect, as the authors have correctly done in the manuscript. My only comment is on the usage of the term cooperativity.

Our response: We thank the reviewer for bringing this to our attention. We agree with the reviewer that this might cause confusion. Therefore, we changed the parameter “ C ” to “ G ” and have removed all the claims on the analogous to cooperativity to make sure there is no link between our parameter to cooperativity (see changes below, in red):

“The parameter $G = 4\mu^2/\kappa_b\kappa$, ~~analogous to the cooperativity in cavity quantum electrodynamics~~, is used to qualify the strength of the mirror ($G \sim 200 - 4300$ in this work. See supplementary materials for details)”

2) Naively, one expects a frequency walk-off across different modes due to group velocity dispersion. This makes the “cold” frequency resonances non-equidistant. It would be helpful to have a discussion on this.

Our response: We thank the reviewer for bringing up this good point. Indeed, the group velocity dispersion can lead to a gradually accumulated frequency detuning between the actual frequency comb line and its corresponding resonance when light is spread along the frequency dimension. This eventually leads to a “soft” frequency “boundary” when the detuning become large. However, our system generates several hundreds of lattice points without touching the dispersion “boundary” (for example, please see Fig. 2 in reference [Hu et al. arXiv:2111.14743 (2021)]). This will not lead to an abrupt discontinuity i.e. it is a “soft boundary”, compared to our current mirrors which has a sharp break on translation symmetry, and therefore provides a strong reflection. Finally, this “boundary” does not serve exactly as a frequency mirror as it will introduce Bloch oscillation [Yuan 3, 1014 (2016)] and this has also been observed in our TFLN system by applying a microwave detuning to effectively create large amounts of dispersion [Hu et al. Optica 7, 1189 (2020)].

We have added a separate paragraph before the summary paragraph to discuss this (see below in red):

“Note that group velocity dispersion can lead to a gradually accumulated frequency detuning between the frequency comb line and the corresponding resonance frequency. Therefore, light that is spread over the frequency domain can eventually reach a “soft” frequency “boundary” when detuning becomes large. However, our platform [32] can generate several hundreds of lattice points without reaching this “soft” dispersion “boundary”. Since this “soft boundary” does not result in a sharp discontinuity of the propagation, and introduces Bloch oscillations [33], it does not serve exactly as a frequency mirror. This effect has been previously observed in our TFLN system by applying a microwave detuning to effectively create large dispersion [18].”

3) The authors make a note in the paragraph starting from line 158 that the coupling strength between the two cavities presented in Fig. 4 can be stronger than the FSR of the cavity1, leading to the presented non-conventional gradual reduction of the FSR of that cavity. They may quantitatively further discuss the relationship between the degree of coupling and observed gradual reduction. They may also wish to provide further supplementary discussion on cases where the coupling strength is weaker or equal to the FSR1.

Our response: We thank the reviewer for helping us improve the manuscript. We have made a more quantitative discussion of how the minimum FSR (in the region of non-conventional gradual reduction of FSR) changes with varied coupling μ , as shown in a new supplementary figure.

Furthermore, we fixed one typo regarding the evaluation of whether it is a two-mode splitting problem or multi-hybrid mode problem (gradual reduced FSR). This is not determined by $\mu > \text{FSR}$ but should be $2\mu > \text{FSR}$ since the conventional degenerate coupling leads to a splitting of 2μ not μ . Therefore, the qualification of the system is not μ is weaker/equal/larger than FSR1 but rather should be compared between 2μ and FSR1.

As a result, we have shown in Fig. S1 how the minimum FSR varies when μ gradually increased to a value comparable to FSR1. As expected, there should not be a sharp phase transition since the increase of μ just gradually generates dispersive coupling to other non-degenerate modes.

We also have also extracted μ in our coupled resonator system and verified $\mu = 6.8$ GHz which is indeed in the regime $2\mu > \text{FSR1}$. We included a simulated version of Fig. 4d to verify it.

We attached all of the above results here (see below in red):

Fig. S1. Relationship between coupling strength and the minimum FSR in the region of gradually reduced FSR. a, Simulated minimum FSR when varying the coupling μ . The phenomenon of gradually reduced FSR happens when the coupling μ starts to become comparable with the FSR of the cavity 1. For example, when the coupling μ is small, it can only couple degenerate modes therefore provide a conventional two mode splitting with a splitting equal to 2μ . When the coupling μ becomes stronger, such that 2μ is larger than the FSR, dispersive coupling starts to contribute, and multi-hybrid modes are formed. We extract the coupling strength μ in

our coupled system based on simulation and obtained $\mu = 6.8$ GHz, which satisfies $2\mu > \text{FSR}$. When the coupling μ becomes extremely large, the minimum FSR will reach its original FSR (10.5 GHz in our simulation here). **b**, Simulated FSR as a function of resonance frequency (Fig. 4d) for the coupling extracted from our system ($\mu = 6.8$ GHz).

4) Further upon the previous comment, they may wish to add general quantitative characterization parameters of their system, including waveguide-ring coupling efficiency and associate losses.

Our response: We thank the reviewer for helping us improve the manuscript. We have included a table in our manuscript, which includes all the extracted parameters in this work (simulation in Fig. 2, polarization system in Fig.3, coupled-resonator system in Fig. 4, and outlook for future improved quality factors of the cavity). Please see below in red:

Table S1 | Parameters of the systems

Parameters	Simulation in Fig. 2	Polarization mirror in Fig. 3	Coupled-resonator mirror in Fig. 4	Outlook
Waveguide-ring coupling κ_e (MHz)	48	20	39	2
Intrinsic loss rate κ_i (MHz)	96	85	169	19
Resonance linewidth κ (MHz)	144	105	207	21
Intrinsic Q	2.0×10^6	2.7×10^6	1.1×10^6	10×10^6
Loaded Q	1.3×10^6	2.2×10^6	0.89×10^6	9.2×10^6
Coupling strength Ω (GHz)	6	6	6	6
Parameter $u = \kappa/\Omega$	2.4×10^{-2}	1.8×10^{-2}	3.5×10^{-2}	0.35×10^{-2}
Propagation loss L (dB/lattice)	0.1	0.076	0.15	0.015
Pump wavelength (nm)	1550	1309	1628	1550
Mirror coupling strength μ (GHz)	2	0.86	6.8	5
Free spectral range FSR (GHz)	10	10.5	10.5	10
Parameter G	772	268	4296	250627
Reflectivity	0.994	0.94	0.999914	0.999997
Power cut off, simulation (dB)	25.9	15.2	44	58.9

Power cut off, experiment (dB)	N/A	16	>30 (limited by noise floor)	N/A
-----	----	---------------------------------	-----

5) In the final paragraph of the manuscript, the authors argue the potential utilization of their system for higher-order topological physics. Evaluating with the recent relevant results reported in the literature (<https://doi.org/10.48550/arXiv.2203.11296>), in which the constructed synthetic boundaries are comparatively sharp, they might consider enriching their discussion of such potential applications so that it more precisely matches their specific type of reported frequency boundaries. A more relevant direction might be the investigation of topological frequency combs (<https://www.nature.com/articles/s41567-021-01302-3>) in such synthetic systems.

Our response: We thank the reviewer for helping us improve our manuscript and bring this paper to our attention. Indeed, our discussion on topological physics can be enriched by discussing topological frequency combs. As a result, we have added the following paragraph (see below in red):

“Specifically, with the ability to introduce defects in frequency domain, the recently emerged topological frequency comb [44] might can be combined with our coupled-resonator platform. TFLN can generate frequency combs using both EO and Kerr nonlinearity simultaneously [32], leading to the realization of a spatial-frequency topological frequency comb.”

As for the boundary type, while we do agree the boundary constructed in [Dutt et al. arXiv: 2203.11296] is pretty sharp, we would like to explain that our TFLN platform actually achieved much stronger boundary than the fiber system. We estimated the following parameters of fiber system: linewidth of the resonance is $\kappa = 0.1277$ MHz, coupling strength Ω due to microwave is $\Omega = 0.642$ MHz, coupling from mirrors $\mu = 0.426$ MHz, parameter $u = 0.2$, propagation loss in frequency domain $L = 0.86$ dB/lattice point, parameter $G = 57$, reflectivity = 0.99 and a power cut-off on mirror = 22.7 dB. Our system parameters are listed in Table 1 and shown in question (4) of the reviewer. Compared to their fiber-coupled-resonator system, our system achieved mirror reflectivity of 0.999914 in TFLN coupled-resonator system while preserving a propagation loss of ~ 0.1 dB/lattice point. Therefore, to our naïve understanding, we believe our mirrors on TFLN should be able to achieve the applications that can be realized in the fiber system.

6)The authors make the claim that their system may be useful for the realization of higher frequency crystals. However, with the significant observed propagation loss, evaluation of the scalability of their system requires more investigation. They may wish to for instance estimate the required microwave driving power to achieve more round trips and multi-trapped states across the frequency crystal.

Our response: We thank the reviewer for helping us improve the manuscript. We have provided a detailed discussion on how to reduce the propagation loss (see the added paragraph in the end of our answer in red) as well as in our added Table (shown when answering previous questions of the reviewer).

Furthermore, although the current propagation loss is still high, our TFLN system already holds the lowest propagation loss that has been achieved in the frequency synthetic dimension. This propagation loss is closely related to the EO frequency comb slope and span. For example, we achieved a slope of 0.7 dB/nm and span of 132 nm in our coupled-resonator EO comb [Hu et al. arXiv: 2111.14743 (2021)] which contains >500 frequency modes, compared to other EO combs such as <100 frequency modes in fiber system [Dutt et al. arXiv: 2203.11296]. Therefore, we believe, coupled-resonator on TFLN is actually the most promising platform that can achieve multi-roundtrip phenomenon provided improvements in quality factors are realized in the future.

We do agree that this propagation loss is still not as low as the loss in spatial-mode propagation. However, the ultimate Q of thin-film lithium niobate has been evaluated as 180 million [Shams-Ansari et al. arXiv: 2203.17133 (2022), APL Photonics (in press), <https://aip.scitation.org/doi/10.1063/5.0095146>]. Using this number, we estimate the propagation loss can be as low as 0.001 dB/lattice point using only ~1-2W microwave power.

As a result, we believe the propagation loss of 0.015 dB/lattice point for 10 million intrinsic Q and the 0.001 dB/lattice point for 180 million intrinsic Q is promising for multi-roundtrips phenomena. We hope the reviewer feels the same too.

Added paragraph:

“Note that the TFLN platform features the lowest propagation loss in the frequency domain to date. The loss in this work (0.076 dB/lattice for the polarization system and 0.15 dB/lattice for the coupled-resonator system) is still higher, however, than the spatial propagation loss of light. Improving the quality factor of our TFLN rings from $\sim 10^6$ in this work to $\sim 10^7$ [35] with the same driving microwave power can reduce the propagation loss in the frequency domain, which is determined by $u = \kappa/\Omega$ (κ is determined by quality factor and Ω is determined by microwave voltage), to 0.015 dB/lattice point, yielding a mirror reflectivity of >0.99999 . Therefore, coupled resonators on TFLN can be promising to investigate multi-roundtrip dynamics in the frequency synthetic dimension, which may lead to the realization of a frequency domain cavity [36].”

7)The sentence starting at line 129 seems to be grammatically incorrect.

Our response: We thank the reviewer for improving our grammar. We changed the sentence to “The first approach that we use to realize frequency domain mirrors is based on polarization mode coupling inside a dispersion-engineered TFLN micro-resonator.”

REVIEWERS' COMMENTS

Reviewer #1 (Remarks to the Author):

The paper is well revised and now it can be accepted for publication.

Reviewer #2 (Remarks to the Author):

Thanks the authors for responding to my comments. I agree with your answers to all my comments. I do not have further comments on the revised manuscript and recommend it for publication in Nature Communications.

Reviewer #3 (Remarks to the Author):

I thank the authors for their detailed response. I believe all my concerns are addressed and recommend the publication in Nat. Comm.

Response to Reviewers' Comments:

Reviewer #1 (Remarks to the Author):

The paper is well revised and now it can be accepted for publication.

Our response: We thank the reviewer for spending their time assessing our work and the support of publication.

Reviewer #2 (Remarks to the Author):

Thanks the authors for responding to my comments. I agree with your answers to all my comments. I do not have further comments on the revised manuscript and recommend it for publication in Nature Communications.

Our response: We thank the reviewer for spending their time assessing our work and the support of publication.

Reviewer #3 (Remarks to the Author):

I thank the authors for their detailed response. I believe all my concerns are addressed and recommend the publication in Nat. Comm.

Our response: We thank the reviewer for spending their time assessing our work and the support of publication.